# Effect of Exogenous Enzymes Cocktail on Performance, Carcass Traits, Biochemical Metabolites, Intestinal Morphology, and Nutrient Digestibility of Broilers Fed Normal and Low-Energy Corn–Soybean Diets

**DOI:** 10.3390/ani12091094

**Published:** 2022-04-23

**Authors:** Abdulmohsen H. Alqhtani, Ali R. Al Sulaiman, Abdulrahman S. Alharthi, Alaeldein M. Abudabos

**Affiliations:** 1Department of Animal Production, College of Food and Agriculture Sciences, King Saud University, P.O. Box 2460, Riyadh 11451, Saudi Arabia; ahalqahtani@ksu.edu.sa (A.H.A.); abalharthi@ksu.edu.sa (A.S.A.); 2National Center for Environmental Technology, Life Science and Environment Research Institute, King Abdulaziz City for Science and Technology, P.O. Box 6086, Riyadh 11442, Saudi Arabia; arsuliman@kacst.edu.sa

**Keywords:** broilers, corn and soybean meal, enzyme cocktail, growth performance, metabolizable energy, nutrient retention

## Abstract

**Simple Summary:**

Corn and soybean meal (SBM) are the principal sources of energy and protein in poultry feed, respectively. Decreasing feed cost per unit of production and increasing the nutritional value of feed ingredients like corn and SBM are continuous concerns for poultry producers. Corn and SBM-based diets are incompletely digested by poultry due to the presence of non-starch polysaccharides (NSPs), which can obstruct the processes of nutrient digestion and absorption. Poultry does not own endogenous enzymes capable of digesting NSPs. The use of enzyme cocktails (EC) and dietary energy sources are essential topics in the poultry industry because they could enhance nutrient utilization by reducing the harmful effects of NSPs. Therefore, evaluating the efficacy of EC that may improve the feeding value of corn–SBM diets warrants attention. The efficacy of two doses of an EC with multiple enzymatic activities at two levels of dietary metabolizable energy (ME) on the performance of broilers fed corn and SBM-based diets up to 35 days was evaluated. Our results indicate that adequately adjusting dietary ME and using the EC with xylanases, β-glucanases, cellulases, proteases, pectinases, and debranching enzymes activities could enhance the nitrogen-corrected apparent ME of corn and SBMbased diets for broiler chickens and potentially offer economic advantages to producers.

**Abstract:**

Ross 308 broilers in a randomized complete block design with a 2 × 2 factorial treatment arrangement (four treatments with 12 replications of six chicks each) were fed corn and SBMbased diets with two concentrations of metabolizable energy (ME) (normal (positive control, PC) and low (negative control, NC)) and two amounts of enzyme cocktail (EC) (0% and 0.005%) for 35 days. Performance, carcass traits, serum metabolites, ileal histology, and apparent nutrient digestibility were evaluated. Compared with the non-supplemented diet, the use of EC improved feed conversion ratio (FCR) over 26–35 and 0–35 days (*p* < 0.01), European performance efficiency factor (EPEF) over 26–35 days (*p* < 0.05), dressing yield (*p* < 0.01), villus height (*p* < 0.05), nitrogen-corrected apparent ME (AMEn) (*p* < 0.01), and serum glucose (*p* < 0.05). Compared with the NC diet, feeding the PC diet improved FCR over all experimental periods (*p* < 0.01, *p* < 0.05, *p* < 0.05, and *p* < 0.01, respectively), EPEF over 0–10 days (*p* < 0.05), and AMEn retention (*p* < 0.01). To conclude, the AMEn of broilers fed corn and SBM diets could be improved by adequately adjusting dietary ME and using a cocktail of non-starch polysaccharide-degrading enzymes, improving commercial benefits to producers.

## 1. Introduction

Growth rate, target body weight, and feed efficiency have been enhanced during recent years [1]; at 30 days of age, 77 g of extra body weight is expected from Ross 308 in 2019 compared to the same strain in 2014 [2]. The fast growth in the poultry industry requires an enormous amount of feed for production. Annually, the poultry compound feed industry accounts for 47% (463 million metric tons) of total feed production globally [3,4]. The continuous increase in feed prices globally is linked to higher demand from developing countries and competition with biofuel energy production [5]. It was estimated that feed costs constitute up to 69%, and the total cost of energy ingredients is about 65–70% of the total costs of feeding in intensive poultry production systems [6,7].

Corn and soybean meal (SBM) make up a substantial percentage of poultry diets because of their nutritional values in terms of energy and protein [8]. Corn is mostly used as cereal grain in the feeds for intensively raised poultry [9], while SBM is the predominant protein source in animal feed universally; it has high protein and amino-acid contents, making it an excellent choice for poultry [10]. The digestibility of a standard corn-SBM diet is limited to approximately 80% for broiler chickens, which is predominantly owing to the existence of insoluble non-starch polysaccharides (NSPs) in corn and SBM [11]. The levels of NSPs in corn and SBM range from 6.8% to 9.4% and from 17% to 30%, respectively [12]. It has been reported that 400 to 450 kcal/kg of digestible energy is undigested because of the NSPs existing in corn and SBM diets [13]. In corn and SBM cells, NSPs are located within the cell-wall matrix, causing a general inhibition of nutrient digestion affecting starch, fat, and protein digestibility, while also correlating closely with apparent metabolizable energy (AME) values [14,15]. It was shown that cell walls containing arabinoxylans act as physical barriers to endogenous enzymes and, therefore, reduce the utilization of starch and protein encapsulated within the endosperm walls [16]. The total quantity of arabinoxylan varies between ingredients but has been estimated to be about 5.2% in corn and 3.3% in SBM [17,18]. Furthermore, SBM contains indigestible oligosaccharides, which mainly consist of α-galactosides that make up 6% of the SBM, including 1.0% raffinose and 4.6% stachyose [19,20]. Oligosaccharides affect the growth performance and health of broilers and are indigestible in poultry due to the absence of endogenous enzymes with α-galactosidase activity [21]. It was estimated that the removal of oligosaccharides from SBM with ethanol led to a 10–15% improvement in metabolizable energy (ME) [22]. Therefore, supplementing exogenous enzymes targeting these indigestible compounds to poultry might improve the access of digestive enzymes to the cell-wall-encapsulated nutrients.

Several studies on the addition of exogenous enzymes to broiler rations have been performed, and enhancements in birds’ growth performance and nutrient availability have been reported. The use of exogenous enzymes can assist in reducing the adverse impacts of NSPs and improve the utilization of dietary nutrients, contributing to improved performance of broilers [23,24,25,26]. Enhancement in broiler’s performance can be correlated with augmented nutrient digestibility and energy utilization [27], in addition to the dietary ME concentration, which has been reported as one of the pivotal factors for the rapid growth in broiler chickens [28]. The addition of an enzyme cocktail (EC) to corn and SBM-based diets to improve the availability of energy for broiler chickens has gained much attention in recent years due to its vital role in the degradation of NSPs in the intestinal tract. Olukosi et al. [29] found that a combination of xylanase, amylase, and protease produced a greater effect for improving retention of energy and protein, as well as increasing the solubilization of NSPs from corn and SBM-based diets given to broilers, than that of protease alone. Amerah et al. [30] noticed positive synergistic effects of exogenous xylanase, amylase, and protease activities on growth performance and AMEn retention in broilers fed corn and SBM-based diets. Stefanello et al. [31] demonstrated that the growth performance, AMEn values, and starch digestibility were improved when broilers were fed corn and SBM diets supplemented with α-amylase and β-xylanase. Another experiment by Saleh et al. [32] concluded that the supplementation of a multienzyme complex in low-ME diets could improve the performance of broilers and decrease the feed cost via upregulation in the mRNA expression of nutrient transporters. On the basis of the abovementioned information, it has been hypothesized that the inclusion of an EC into corn and SBM-based diets formulated with a reduced ME level could enhance the utilization of nutrients and subsequently improve the growth performance of broilers and the economic value of the diets. Although several studies have been executed to study the influence of multienzyme complexes in corn and SBM diets, little knowledge is available concerning the interaction effect of dietary ME and supplemental EC on the productivity of broilers. Therefore, the present study was performed to assess the effectiveness of using an EC with multiple enzymatic activities at various dietary ME concentrations on growth performance, carcass yields, serum biochemical variables, intestinal histological changes, and digestibility of certain nutrients in broilers given corn and SBM diets.

## 2. Materials and Methods

### 2.1. Husbandry and Treatments

A total of 288 day old mixed-sex broilers (Ross 308) with comparable initial body weights (BW) were raised at six chicks/cage in battery brooders with a maximum stocking density of 30 kg/m^2^ in an environmentally controlled room under standard environmental, managerial, and hygienic practices [33] from days 1 to 35. They were acquired from a commercial hatchery and immunized at the hatchery following a vaccination regime for Ross strain. Birds had free access to mash feed and freshwater during the course of the trial. Feeds (Table 1) based on corn and SBM for starter (0–10 days), grower (11–25 days), and finisher (26–35 days) phases were prepared according to Ross broiler nutritional requirements [34] except for ME.

The experiment was performed as a randomized complete block design with four dietary treatments arranged as a 2 × 2 factorial lasting 35 days. Each treatment was replicated 12 times with six chicks each. The treatments were as follows: positive control (PC), normal ME diets for each phase of growth (3000, 3100, and 3200 kcal/kg, respectively) without EC; negative control (NC), the calculated ME values of the PC diets were reduced by 60, 90, and 90 kcal for the three phases of growth, respectively, without adding EC; PC supplemented with 0.005% EC; NC supplemented with 0.005% EC. The exogenous EC (Rovabio Advance, Adisseo France SAS, Antony, France) contains several activities: xylanases (endo-1,4-β-xylanase and β-xylosidase), β-glucanases (endo-1,3(4)-β-glucanase and laminaribase), cellulases (endo-1,4-β-glucanase, β-glucosidase, and cellobiohydrolase), proteases (aspartic protease and metalloprotease), pectinases (pectin esterase, α-galactosidase, endo-1,5-α-arabinase, polygalacturonase, and rhamnogalacturonase), debranching enzymes (α-arabinofuranosidase, ferulic acid esterase, and α-glucuronidase), and others (endo-1,4-β-mannanase and β-manosidase).

### 2.2. Sampling and Measurements

Feed intake (FI), weight gain (WG), feed conversion ratio (FCR), and European performance efficiency factor (EPEF) were computed for the 0–10, 11–25, 26–35, and 0–35 day periods. The FCR was adjusted for mortality and EPEF was determined employing the following equation: ((BW (kg) × livability (%))/(FCR × age (days))) × 100.

On day 35, one bird per replicate with a BW close to the cage average weight was chosen, and the blood for biochemical analyses was taken from the jugular vein and centrifuged at 3000 rpm for 10 min to separate serum that was stored at −80 °C. The serum levels of total protein, albumin, glucose, triglyceride, alanine transaminase, and aspartate aminotransferase were determined by utilizing enzymatic colorimetric kits (M di Europa GmbH Wittekamp 30. D-30163 Hannover, Germany) as specified by the manufacturer. Serum globulin was estimated accordingly by subtracting albumin from total protein.

For carcass and organs yield, chickens (12 birds/treatment) were individually weighed and slaughtered, and each carcass was defeathered, eviscerated, and dressed. The dressing percentage was determined as the proportion of hot carcass weight to the preslaughter live weight. Weights of hot carcass, breast, and legs were taken and expressed as proportions of BW at slaughter. The weight of giblets including abdominal fat, spleen, bursa, liver, and empty gizzard and intestine was also taken separately and expressed as a percentage of live BW at slaughter.

For histological measurements, a sample of the lower ileum was cut (1 cm), washed in physiological saline solution, and fixed in 10% formalin. Serial 5 μm sections of the tissues were prepared and put on glass slides for hematoxylin and eosin staining as previously described by Abudabos et al. [35]. The height and width of at least 10 intact villi were measured utilizing a microscope with a PC-based image analysis system (Olympus NV, Aartselaar, Belgium). The following equation was utilized to compute the total area of villi: (2π) × (villus width/2) × (villus height) [36].

At the end of the growth trial, 12 birds per treatment were kept individually in metabolism cages for the digestibility assessment. For a 72 h period, excreta were collected using the total collection method [37] and pooled within each replication, and FI was recorded. After weighing and drying at 60 °C until constant weight, excreta and feed were ground to pass through a 0.5 mm screen and stored at −4 °C pending analyses. Dry matter (DM), crude protein (CP), and ether extract (EE) contents of the diets and excreta were determined following AOAC [38] standard methods: 930.15 (drying procedure), 984.13 (Kjeldahl procedure), and 920.39 (Soxhlet procedure). Gross energy was also estimated employing a bomb calorimeter (Parr Instrument, Moline, IL, USA), and nitrogen-corrected AME (AMEn) was computed as described by Khalil et al. [39]. The apparent retention of DM, CP, and EE as a percentage was calculated according to Nkukwana et al. [40] with the following formula: (nutrient ingested − nutrient emptied/nutrient ingested) × 100.

### 2.3. Statistical Analysis

The data were analyzed by two-way ANOVA using the GLM procedure of SAS 9.4 (SAS Institute Inc, Cary, NC, USA) to determine the main effects of the EC and ME, as well as the interaction between these two factors. The experimental unit was the cage for performance data and individual bird for other data. Differences between means were established by the Tukey test at a significance level of *p* < 0.05. Results are given as means and pooled SEM.

## 3. Results

### 3.1. Performance Variables

The performance results for the starter period (0–10 days) showed no significant differences in FI, BW, FCR, and EPEF due to EC or ME × EC (*p* > 0.05). Conversely, the PC group had improved FCR (*p* < 0.01) and EPEF (*p* < 0.05) in comparison with the NC group (Table 2). Likewise, FCR for the growing period (11–25 days) was affected by ME; birds converted feed more efficiently in the PC group compared to the NC group (*p* < 0.05). However, EC or ME × EC had no effects (*p* > 0.05) on any of the measured variables during that period (Table 2).

The data on the finisher period (26–35 days) revealed significant effects for the ME and EC on FCR; birds converted feed more efficiently in the PC (*p* < 0.05) and EC (*p* < 0.01) groups compared to the NC and non-supplemented groups, respectively. Similarly, EPEF improved (*p* < 0.05) as a result of EC addition over the non-supplemented diet. However, no interaction (*p* > 0.05) between ME and EC was found for any variable measured during that period (Table 3). The result for the overall period (0–35 days) showed significant effects for EC and ME on FCR (*p* < 0.01); birds that received the PC diet had a better FCR as compared to those fed the NC diet (1.41 vs. 1.49 g:g, respectively) and EC improved FCR compared to the non-supplemented diet (1.43 vs. 1.47 g:g, respectively), with no interaction (*p* > 0.05) between the two factors for any measure (Table 3).

### 3.2. Carcass Measurements

There was no interaction between ME × EC for any of the measurements (*p* > 0.05). The mean percentage of dressing yield and parts yield of broilers at 35 days showed that EC increased dressing yield (*p* < 0.01) by 1.3%, while feeding a low-ME diet lowered abdominal fat (*p* < 0.05). The weight percentages of gizzard and intestine were lower (*p* < 0.05) for the EC group compared to the non-supplemented group (Table 4).

### 3.3. Intestinal Histomorphometry and Digestibility of Nutrients

The interaction of ME and EC did not affect ileum histology at 35 days and the apparent digestibility of nutrients over 35–38 days (*p* > 0.05). Villus height was increased as a result of EC (*p* < 0.05). On the other hand, the ME level did not affect the height, width, or total area of villi (*p* > 0.05). The retention of AMEn was influenced by dietary ME and EC inclusion (*p* < 0.01); birds that received the PC and EC diets retained more AMEn when compared to the NC and non-supplemented diets (3165 vs. 3073 kcal/kg and 3130 vs. 3108 kcal/kg, respectively) (Table 5).

### 3.4. Blood Biochemical Indexes

Serum biochemical variables of broiler at 35 days are presented in Table 6. Serum total protein, albumin, globulin, alanine transaminase, and aspartate aminotransferase were similar across all groups and were not influenced by EC, ME, or their interaction (*p* > 0.05). However, glucose level was the only variable affected by EC (*p* < 0.05); higher glucose concentration was found in birds that received the diet with EC (241 mg/dL) compared to those with the non-supplemented diet (225 mg/dL). Additionally, triglyceride concentration was affected by dietary ME level (*p* < 0.01); birds given the PC diet had a higher triglyceride when compared to those fed the NC diet (54.8 vs. 48.6, respectively).

## 4. Discussion

Broilers show an amazing ability to control the intake of energy through modifying their FI as the diet energy concentration changes [41]. In the present study, FI and BWG were insignificantly affected by lowering dietary ME by 60, 90, and 90 kcal/kg for starter, grower, and finisher phases, respectively, indicating that ME of the NC group was not low enough to cause a significant difference. Other studies used lower ME levels (132, 133, and 270 kcal/kg) for the three rearing phases, respectively, and reported significant differences in growth rate and feed efficiency [42,43,44]. Nevertheless, FCR was negatively impacted by the NC diet for the overall period (5.3% reduction) in comparison with the PC diet, demonstrating that broilers were incapable of meeting their dietary ME requirements to maximize FCR. Similarly, Masey O’Neill et al. [45] reported negative effects on FCR when reducing dietary ME through 42 days. Decreasing dietary ME in the NC diet did not affect processing variables herein except for fat. Williams et al. [44] reported that a 132 kcal reduction in ME in the NC diet caused a reduction in carcass yields including fat pad.

The exogenous enzyme preparation used in the current study contains multiple active substances. It was suggested that the exogenous EC with several enzymatic activities can target multiple components of feed, certainly having a greater effect than individual enzymes, which target one substrate [30,43,44,46]. Improvements in BW and FCR were observed with the addition of a cocktail of non-starch polysaccharide-degrading enzymes (NSPase); Slominski [47] reported 3.9% and 3.2% improvements in BWG and FCR of broilers receiving corn and SBM diets with NSPase. Similar enhancements in BW and FCR were found in low-energy broiler feeds supplemented with NSPase [30,43,44,48]. Our results are in partial agreement with the aforementioned reports and reported a significant improvement in FCR by 3.2% for the overall period when EC was supplemented. On the other hand, the supplemental EC did not affect WG during the whole experimental period, a result which disagrees with several studies that reported a greater WG in broilers fed diets with multienzyme mixtures [49,50,51,52]. Feed consumption of broilers given the EC-supplemented diets was similar to those fed the non-supplemented diets during all periods, which disagrees with the finding of Ranade and Rajmane [53] who reported that broilers lowered their FI when diets contained α-amylase, xylanase, cellulase, protease, and β-glucanase activities. Furthermore, EC supplementation increased the dressing percentage by 1.3% without any effect on breast and leg yields; similarly, Farran et al. [54] reported that enzyme preparations had no significant effects on pectoralis muscle major and leg yields.

The retention of AMEn in the current study was augmented as a result of supplementing the diet with EC or formulating the diet with normal ME levels. The report of Amerah et al. [30] is in agreement with the results obtained herein, where AMEn values were improved in broilers fed corn and SBM-based diets supplemented with exogenous xylanase, amylase, and protease as combined activities, with a nutritionally adequate diet having the highest AMEn mean value. Similarly, Graham et al. [55] reported that the true ME of a feed that contained SBM treated with α-galactosidase was increased by 354 kcal/kg due to degradation of raffinose and stachyose in SBM by 69% and 54%, respectively. Ao et al. [56] also showed that α-galactosidase augmented the AMEn values of corn and SBM-based diets. Other researchers reported an improvement in digestible energy by 3.2% and 5.3% in broilers fed corn and SBM diets supplemented with xylanase [57,58]. Xylanase is used to break down NSPs and reduce the viscosity of digesta to improve nutrient digestibility [59,60]. The improvements in AMEn reported herein could happen as a direct effect of enzymes such as pectin esterase and β-l-arabinofuranosidase, which were part of the cocktail used in the current experiment. Pectin esterase and β-l-arabinofuranosidase are required to hydrolyze the backbone of arabinoxylans and pectins since they are not reachable for enzymes such as endo-β-1, 4-xylanases because they are highly branched [61]. Moreover, the improvements in AMEn could be related to the action of cellobiohydrolase and β-glucosidase in cellulose degradation; both enzymes aid endo-β-glucanases for the random cleavage of β-1, 4 linkages in the glucan chains which form the cellulose microfibrils [62].

The intestinal morphological structure reflected by the villus height and total area is one of the major indicators of animal health and production [63]. In the present study, supplementation with EC resulted in a significant increase in the villus height of the ileum, suggesting greater absorption of available nutrients. This is consistent with the results of Karimi et al. [64], who found that ileal villus height was significantly increased by dietary β-mannanase and β-glucanase enzymes. Similar findings were reported in broiler chickens fed with exogenous xylanase and β-glucanase enzymes [65]. The improvement of intestinal morphology with the EC could be attributed to ameliorating the harmful impacts of the NSP content of the diet on the intestinal villi.

## 5. Conclusions

The findings of the present research illustrated that adding a cocktail of enzymes comprising xylanases, β-glucanases, cellulases, proteases, pectinases, debranching enzymes, and other activities to corn and SBM-based diets improved FCR for the finisher and overall periods, EPEF for the finisher period, dressing yield, VH, AMEn retention, and glucose metabolism of broilers raised to 35 days of age. In addition, formulating the corn-SBM broiler rations with adequate levels of ME decreased FCR throughout the entire growth period and increased AMEn retention. The advantageous influence of the EC could be linked to the hydrolysis of the different components of NSPs found in corn and SBM, which possibly results in economic benefits for industrial broiler production.

## Figures and Tables

**Table 1 animals-12-01094-t001:** Ingredients and nutrient contents of the experimental diets (%, as-fed basis).

Ingredients	Starter		Grower		Finisher	
	PC ^1^	NC	PC	NC	PC	NC
Yellow corn	51.6	51.6	58.5	58.8	59.8	60.6
Soybean meal	32.4	32.4	28.2	28.2	27.0	27.0
Corn oil	3.30	2.50	3.60	2.48	4.34	3.10
Corn gluten meal	6.30	6.10	4.71	4.30	5.10	4.50
Wheat bran	2.00	3.00	1.00	2.20	0.00	1.10
Dicalcium phosphate	2.05	2.05	1.82	1.82	1.68	1.67
Ground limestone	0.90	0.90	0.88	0.88	0.87	0.87
Choline chloride	0.05	0.05	0.00	0.00	0.00	0.00
dl-Methionine	0.30	0.30	0.26	0.26	0.25	0.25
l-Lysine	0.38	0.38	0.30	0.33	0.26	0.26
Salt	0.40	0.40	0.40	0.40	0.40	0.40
Threonine	0.14	0.14	0.13	0.13	0.08	0.08
Vitamin–mineral premix ^2^	0.20	0.20	0.20	0.20	0.20	0.20
Calculation of nutrients						
Metabolizable energy ^3^	3000	2940	3100	3010	3200	3110
Crude protein	23.0	23.0	21.5	21.5	20.0	20.0
Available phosphorus	0.48	0.48	0.44	0.44	0.41	0.41
Calcium	0.96	0.96	0.87	0.87	0.81	0.81
Lysine	1.28	1.28	1.15	1.15	1.06	1.06
Methionine + cysteine	0.95	0.95	0.85	0.85	0.83	0.83
Threonine	0.86	0.86	0.77	0.77	0.71	0.71

^1^ PC, positive control; NC, negative control. ^2^ Vitamin–mineral premix supplied per kg diet: vitamin A, 12,000,000 IU; vitamin D3, 5,000,000 IU; vitamin E, 80,000 IU; vitamin K3, 3200 mg; vitamin B1, 3200 mg; vitamin B2, 8600 mg; vitamin B3, 65,000 mg; vitamin B5, 20,000 mg; vitamin B6, 4300 mg; vitamin B7, 220 mg; vitamin B9, 2200 mg; vitamin B12, 17 mg; antioxidant (butylated hydroxyanisole + butylated hydroxytoluene), 50,000 mg; copper, 16,000 mg; iodine, 1250 mg; iron, 20,000 mg; manganese, 120,000 mg; selenium, 300 mg; zinc, 110,000 mg. ^3^ Expressed in kcal/kg. The EC was supplemented to the basal diets at a level of 0.005%.

**Table 2 animals-12-01094-t002:** Effect of dietary metabolizable energy (ME) level, enzyme cocktail (EC) supplementation, and their interaction on broiler performance in the starter (0–10 days) and grower (11–25 days) phases.

Treatments ^1^		Response Variables ^2^							
		0–10 Days				11–25 Days			
ME	EC (%)	FI	WG	FCR	EPEF	FI	WG	FCR	EPEF
		(g)	(g)	(g:g)	(%)	(g)	(g)	(g:g)	(%)
PC	0	179	164	1.10	168	1083	808	1.35	310
NC	0	178	156	1.14	146	1096	752	1.47	271
PC	0.005	179	165	1.08	168	1092	800	1.37	297
NC	0.005	175	157	1.12	157	1137	814	1.40	295
SEM ^3^		5.45	5.90	0.015	7.62	20.7	26.3	0.028	12.7
Main effects									
ME level									
PC		179	164	1.09 ^b^	168 ^a^	1087	804	1.36 ^b^	303
NC		177	156	1.13 ^a^	152 ^b^	1117	783	1.43 ^a^	283
EC (%)									
0		179	160	1.12	157	1089	780	1.41	291
0.005		177	161	1.10	162	1115	807	1.39	296
Significance									
ME		NS	NS	0.01	0.05	NS	NS	0.05	NS
EC		NS	NS	NS	NS	NS	NS	NS	NS
ME × EC		NS	NS	NS	NS	NS	NS	NS	NS

^1^ PC (positive control), Aviagen dietary ME recommendations; NC (negative control), lower levels of dietary ME (by 60 and 90 kcal/kg for the starter and grower/finisher rations, respectively). ^2^ EPEF, European performance efficiency factor; FCR, feed conversion ratio; FI, feed intake; WG, weight gain. ^3^ SEM, pooled standard error of the mean. ^ab^ Means in the same column with different superscript letters differ (*p* < 0.05). NS, not significant.

**Table 3 animals-12-01094-t003:** Effect of dietary metabolizable energy (ME) level, enzyme cocktail (EC) supplementation, and their interaction on broiler performance in the finisher (26–35 days) and overall (0–35 days) phases.

Treatments ^1^		Response Variables ^2^								
		26–35 Days				0–35 Days				
ME	EC (%)	FI	WG	FCR	EPEF	FI	WG	FCR	EPEF	FBW
		(g)	(g)	(g:g)	(%)	(g)	(g)	(g:g)	(%)	(kg)
PC	0	1285	819	1.58	335	2547	1790	1.43	370	1.84
NC	0	1335	815	1.65	316	2609	1723	1.52	342	1.81
PC	0.005	1284	884	1.46	369	2555	1828	1.40	384	1.87
NC	0.005	1289	837	1.54	343	2602	1793	1.45	363	1.84
SEM ^3^		28.4	27.9	0.036	13.7	42.7	42.2	0.020	13.1	0.049
Main effects										
ME level										
PC		1282	851	1.52 ^b^	352	2551	1809	1.41 ^b^	377	1.86
NC		1313	825	1.60 ^a^	330	2605	1758	1.49 ^a^	353	1.83
EC (%)										
0		1310	860	1.61 ^a^	326 ^b^	2577	1756	1.47 ^a^	356	1.83
0.005		1286	817	1.50 ^b^	356 ^a^	2578	1810	1.43 ^b^	373	1.86
Significance										
ME		NS	NS	0.05	NS	NS	NS	0.01	NS	NS
EC		NS	NS	0.01	0.05	NS	NS	0.01	NS	NS
ME × EC		NS	NS	NS	NS	NS	NS	NS	NS	NS

^1^ PC (positive control), Aviagen dietary ME recommendations; NC (negative control), lower levels of dietary ME (by 60 and 90 kcal/kg for the starter and grower/finisher rations, respectively). ^2^ EPEF, European performance efficiency factor; FBW, final body weight; FCR, feed conversion ratio; FI, feed intake; WG, weight gain. ^3^ SEM, pooled standard error of the mean. ^ab^ Means in the same column with different superscript letters differ (*p* < 0.05). NS, not significant.

**Table 4 animals-12-01094-t004:** Effect of dietary metabolizable energy (ME) level, enzyme cocktail (EC) supplementation, and their interaction on broiler processing variables (%, based on preslaughter weights) at 35 days.

Treatments ^1^		Response Variables								
ME	EC (%)	Dressing	Breast	Leg	Fat	Spleen	Bursa	Liver	Gizzard	Intestine
PC	0	70.9	26.7	21.8	1.20	0.117	0.138	3.38	2.34	2.78
NC	0	70.3	25.4	22.1	0.84	0.112	0.153	3.29	2.49	2.85
PC	0.005	72.2	27.6	22.5	0.98	0.108	0.129	3.32	2.50	2.75
NC	0.005	71.7	27.2	22.5	0.78	0.106	0.166	3.29	2.49	2.75
SEM ^2^		0.35	0.65	0.82	0.13	0.075	0.044	0.079	0.17	0.12
Main effects										
ME level										
PC		71.6	27.2	22.1	1.09 ^a^	0.107	0.143	3.32	2.42	2.76
NC		70.9	26.3	22.3	0.81 ^b^	0.112	0.149	3.33	2.46	2.71
EC (%)										
0		70.6 ^b^	26.1	21.9	1.00	0.115	0.084	2.08	2.59 ^a^	3.15 ^a^
0.005		71.9 ^a^	27.3	22.5	0.88	0.182	0.126	2.03	2.18 ^b^	2.87 ^b^
Significance										
ME		NS	NS	NS	0.05	NS	NS	NS	NS	NS
EC		0.01	NS	NS	NS	NS	NS	NS	0.05	0.05
ME × EC		NS	NS	NS	NS	NS	NS	NS	NS	NS

^1^ PC (positive control), Aviagen dietary ME recommendations; NC (negative control), lower levels of dietary ME (by 60 and 90 kcal/kg for the starter and grower/finisher rations, respectively). ^2^ SEM, pooled standard error of the mean (*n* = 12). ^ab^ Means in the same column with different superscript letters differ (*p* < 0.05). NS, not significant.

**Table 5 animals-12-01094-t005:** Effect of dietary metabolizable energy (ME) level, enzyme cocktail (EC) supplementation, and their interaction on ileal morphology at 35 days and apparent retention of nutrients in broilers over 35–38 days.

Treatments ^1^		Response Variables ^2^						
ME	EC (%)	VH	VW	VTA	DM	CP	EE	AMEn
		(µm)	(µm)	(µm^2^)	(%)	(%)	(%)	(kcal/kg)
PC	0	504	92.5	0.147	78.1	66.8	80.6	3149
NC	0	512	88.1	0.132	77.8	65.9	80.1	3067
PC	0.005	547	89.8	0.151	78.5	67.0	80.7	3180
NC	0.005	529	81.9	0.151	77.9	66.2	80.6	3079
SEM ^3^		7.80	4.87	0.007	0.22	0.56	0.35	5.52
Main effects								
ME level								
PC		526	90.2	0.141	78.2	66.9	80.7	3165 ^a^
NC		521	86.7	0.148	77.8	66.1	80.3	3073 ^b^
EC (%)								
0		508 ^b^	87.2	0.139	78.2	66.4	80.6	3108 ^b^
0.005		538 ^a^	89.8	0.151	77.9	66.6	80.3	3130 ^a^
Significance								
ME		NS	NS	NS	NS	NS	NS	0.01
EC		0.05	NS	NS	NS	NS	NS	0.01
ME × EC		NS	NS	NS	NS	NS	NS	NS

^1^ PC (positive control), Aviagen dietary ME recommendations; NC (negative control), lower levels of dietary ME (by 60 and 90 kcal/kg for the starter and grower/finisher rations, respectively). ^2^ AMEn, nitrogen-corrected apparent ME; CP, crude protein; DM, dry matter; EE, ether extract; VH, villus height; VTA, villus total area; VW, villus width. ^3^ SEM, pooled standard error of the mean (*n* = 12). ^ab^ Means in the same column with different superscript letters differ (*p* < 0.05). NS, not significant.

**Table 6 animals-12-01094-t006:** Effect of dietary metabolizable energy (ME) level, enzyme cocktail (EC) supplementation, and their interaction on serum biochemical indices of broilers at 35 days.

Treatments ^1^		Response Variables ^2^						
ME	EC (%)	TP	ALB	GLO	GLU	TG	ALT	AST
		(g/dL)	(g/dL)	(g/dL)	(mg/dL)	(mg/dL)	(IU/L)	(IU/L)
PC	0	2.72	1.52	1.19	227	51.1	18.4	306
NC	0	2.60	1.52	1.08	224	48.7	19.9	290
PC	0.005	2.86	1.57	0.99	248	58.6	23.4	265
NC	0.005	2.74	1.57	1.06	235	48.5	22.0	311
SEM ^3^		0.12	0.08	0.12	6.58	2.61	3.55	18.7
Main effects								
ME level								
PC		2.79	1.55	1.10	237	54.8 ^a^	20.9	286
NC		2.67	1.55	1.10	229	48.6 ^b^	20.9	301
EC (%)								
0		2.66	1.52	1.14	225 ^b^	49.9	19.2	298
0.005		2.80	1.57	1.03	241 ^a^	53.5	22.7	288
Significance								
ME		NS	NS	NS	NS	0.01	NS	NS
EC		NS	NS	NS	0.05	NS	NS	NS
ME × EC		NS	NS	NS	NS	NS	NS	NS

^1^ PC (positive control), Aviagen dietary ME recommendations; NC (negative control), lower levels of dietary ME (bu 60 and 90 kcal/kg for the starter and grower/finisher rations, respectively). ^2^ TP, total protein; ALB, albumin; GLO, globulin; GLU, glucose; TG, triglyceride; ALT, alanine transaminase; AST, aspartate aminotransferase. ^3^ SEM, pooled standard error of the mean (*n* = 12). ^ab^ Means in the same column with different superscript letters differ (*p* < 0.05). NS, not significant.

## Data Availability

The data that support the findings of this study are available from the corresponding author upon reasonable request.

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
