# Peer review of "Effect of Exogenous Enzymes Cocktail on Performance, Carcass Traits, Biochemical Metabolites, Intestinal Morphology, and Nutrient Digestibility of Broilers Fed Normal and Low-Energy Corn–Soybean Diets"

_animals, 2022, doi:10.3390/ani12091094_

Round 1

Reviewer 1 Report

The authors have conducted a study on the effects of an enzyme cocktail in broilers. Please consider the following comments:

Comment 1: The Introduction lacks two main things: (1) The background information does not support why this study was convenient to be conducted. The design of this study does not provide any means to clarify the supposedly inconsistent reports regarding a possible synergistic effect when combined enzymes are fed as a cocktail. That point was stated (clarifying the inconsistent results) to be the lacking information to be covered by the study. (2) The Introduction does not provide any means to show the novelty of the information provided. Please redesign the Introduction accordingly.

Comment 2: After correcting the Introduction, update the Discussion (and Conclusion, if convenient) section. Please indicate what lines have been modified.

Comment 3: Line 26 should say "two doses of an EC with..."

Comment 4: Please clarify what "dressing" means for you in the context of your study.

Comment 5: Shouldn't line 45 say "NC diet decreased EPEF".

Comment 6: Shouldn't line 47 say "and increased FCR".

Comment 7: Please double-check the whole manuscript according to comments 4 and 5.

Comment 8: Please clarify how the diets were offered (mash, pellet).

Comment 9: Please replace "parameters" with "response variables" or "variables" in the whole manuscript.

Comment 10: Please ensure in the whole manuscript it says "kg", not "Kg".

Author Response

Reviewer 1

Comment 1: The Introduction lacks two main things: (1) The background information does not support why this study was convenient to be conducted. The design of this study does not provide any means to clarify the supposedly inconsistent reports regarding a possible synergistic effect when combined enzymes are fed as a cocktail. That point was stated (clarifying the inconsistent results) to be the lacking information to be covered by the study. (2) The Introduction does not provide any means to show the novelty of the information provided. Please redesign the Introduction accordingly.

Done as requested

Comment 2: After correcting the Introduction, update the Discussion (and Conclusion, if convenient) section. Please indicate what lines have been modified.

Done as requested

Comment 3: Line 26 should say "two doses of an EC with..."

Done as requested

Comment 4: Please clarify what "dressing" means for you in the context of your study.

The dressing percentage or yield was determined as the proportion of hot carcass weight to the pre-slaughter live weight.

Comment 5: Shouldn't line 45 say "NC diet decreased EPEF".

Done as requested

Comment 6: Shouldn't line 47 say "and increased FCR".

Done as requested

Comment 7: Please double-check the whole manuscript according to comments 5 and 6.

Done as requested

Comment 8: Please clarify how the diets were offered (mash, pellet).

Done as requested

Comment 9: Please replace "parameters" with "response variables" or "variables" in the whole manuscript.

Done as requested

Comment 10: Please ensure in the whole manuscript it says "kg", not "Kg".

Done as requested

Reviewer 2 Report

Nice work, will add a new addition to the existing data.

I have got an inquiry from the authors.

Could you please justify selecting the Corn-SBM diet while using Enzyme Cocktail (EC)? The nutrients in corn and SBM are generally highly digestible. Corn is a non-viscous grain with low soluble NSP. Soluble NSP is a major concern of the poultry industry because of reducing digestion and absorption of nutrients. Again, most of the antinutrients in SBM are deactivated during heat processing. When you are using NSP degrading enzymes/EC, probably you could observe more benefits and differences in performance, FCR using the alternative grains with higher soluble NSP (wheat, barley).

Anyway, my comments and corrections are as follows: 

LN 15, 32: Follow the journal format, the summary and abstract should be no more than 200 words in each

LN 66-67: ‘Mostly used cereal’

LN 105: ‘day-old’ replacing ‘1-day-old’

LN 114: Suggest adding the percentage of enzyme cocktail (EC) within the diet composition in Table 1/as a footnote.

LN 126-127: Not clear, I believe it should be PC supplemented with 0.005% EC; and NC supplemented with 0.005% EC

LN 191-192: Arrange the abbreviations alphabetically (i.e. EPEF, FCR, FI, WG)

LN 209-210: Arrange the abbreviations alphabetically

LN 237-238: Arrange the abbreviations alphabetically

LN 246, 247: just write 241 and 225 mg/dl (to keep the values same as Table)

LN 276: Ignore insignificant results

Discussion: There is no discussion on Histomorphometry data, consider adding some lines

Reference (Re-check all the references following format)

References No. 1,3,4: DOI not needed during submission. Follow the guideline. Correct all the references where applicable.

Reference No. 7, 8,22, 25, 29, 40, 44, 50 (the highlighted words should start with a small letter to keep consistency throughout the whole reference section)

LN 384: Re-check the name

LN 385: ‘MSc Thesis’ replacing ‘Dissertation’

LN 388: Biotech.

Author Response

Reviewer 2

Could you please justify selecting the Corn-SBM diet while using Enzyme Cocktail (EC)? The nutrients in corn and SBM are generally highly digestible. Corn is a non-viscous grain with low soluble NSP. Soluble NSP is a major concern of the poultry industry because of reducing digestion and absorption of nutrients. Again, most of the antinutrients in SBM are deactivated during heat processing. When you are using NSP degrading enzymes/EC, probably you could observe more benefits and differences in performance, FCR using the alternative grains with higher soluble NSP (wheat, barley).

I agree with you, usually exogenous enzymes works better with barley and wheat based diets. However, research showed that the total quantity of arabinoxylan has been estimated to be about 5.2% in corn and 3.3% in SBM [17,18]. Moreover, this was part of bigger project, where we used enzyme with typical corn-soy diets to show that there is room for improvements. Other experiments we conducted were based on different ingredients and some local by-products and the enzyme showed significant improvements.

LN 15, 32: Follow the journal format, the summary and abstract should be no more than 200 words in each

Done as requested

LN 66-67: ‘Mostly used cereal’

Done as requested

LN 105: ‘day-old’ replacing ‘1-day-old’

Done as requested

LN 114: Suggest adding the percentage of enzyme cocktail (EC) within the diet composition in Table 1/as a footnote.

Done as requested

LN 126-127: Not clear, I believe it should be PC supplemented with 0.005% EC; and NC supplemented with 0.005% EC

Done as requested

LN 191-192: Arrange the abbreviations alphabetically (i.e. EPEF, FCR, FI, WG)

Done as requested

LN 209-210: Arrange the abbreviations alphabetically

Done as requested

LN 237-238: Arrange the abbreviations alphabetically

Done as requested

LN 246, 247: just write 241 and 225 mg/dl (to keep the values same as Table)

Done as requested

LN 276: Ignore insignificant results

Done as requested

Discussion: There is no discussion on Histomorphometry data, consider adding some lines

Done as requested

Reference (Re-check all the references following format)

References No. 1,3,4: DOI not needed during submission. Follow the guideline. Correct all the references where applicable.

References were formulated using Mendeley Reference Manager with Animals format.

Reference No. 7, 8,22, 25, 29, 40, 44, 50 (the highlighted words should start with a small letter to keep consistency throughout the whole reference section)

Done as requested

LN 384: Re-check the name

Done as requested

LN 385: ‘MSc Thesis’ replacing ‘Dissertation’

Done as requested

LN 488: Biotech.

3 Biotech is a hybrid journal, Springer.

Round 2

Reviewer 1 Report

Well done.

Reviewer 2 Report

The comments are addressed properly.